# Structural diversity of oligomeric β-propellers with different numbers of identical blades

Evgenia Afanasieva[†], Indronil Chaudhuri[†‡], Jörg Martin[†], Eva Hertle, Astrid Ursinus, Vikram Alva, Marcus D Hartmann, Andrei N Lupas*

Department of Protein Evolution, Max Planck Institute for Developmental Biology, Tübingen, Germany

**Abstract** β-Propellers arise through the amplification of a supersecondary structure element called a blade. This process produces toroids of between four and twelve repeats, which are almost always arranged sequentially in a single polypeptide chain. We found that new propellers evolve continuously by amplification from single blades. We therefore investigated whether such nascent propellers can fold as homo-oligomers before they have been fully amplified within a single chain. One- to six-bladed building blocks derived from two seven-bladed WD40 propellers yielded stable homo-oligomers with six to nine blades, depending on the size of the building block. High-resolution structures for tetramers of two blades, trimers of three blades, and dimers of four and five blades, respectively, show structurally diverse propellers and include a novel fold, highlighting the inherent flexibility of the WD40 blade. Our data support the hypothesis that subdomain-sized fragments can provide structural versatility in the evolution of new proteins.
DOI: https://doi.org/10.7554/eLife.49853.001

*For correspondence:
andrei.lupas@tuebingen.mpg.de

[†]These authors contributed equally to this work

Present address: [‡]Carl Zeiss Vision GmbH, Aalen, Germany

## Introduction

Current evolutionary scenarios generally treat domains as the unit of protein evolution. Domains, however, already display a level of complexity that seems to preclude their origin by chemical processes in an abiotic environment. We have proposed that the first domains evolved from a pool of peptides with the propensity to form supersecondary structures, which originated in the context of RNA-based replication and catalysis (*Lupas et al., 2001*; *Söding and Lupas, 2003*). Initially, these peptides were entirely dependent on an RNA scaffold for their structure and activity, but increasing complexity allowed them to form structures by excluding water through hydrophobic contacts and thus gain independence from RNA; in this model, protein folding was an emergent property of peptide-RNA coevolution (*Lupas and Alva, 2017*). Using a computational approach, we have retraced 40 of these peptides, whose presence in different folds suggests that they predated the first folded proteins (*Alva et al., 2015*). The most prolific, such as the nucleotide-binding helix-turn-helix motif, found in 14 different folds (*Sauer et al., 1982*; *Steitz et al., 1982*; *Rosinski and Atchley, 1999*; *Aravind et al., 2005*), or the (di)nucleotide-binding β-α-β Rossmann motif, found in 10 different folds (*Rossmann et al., 1974*; *Dym and Eisenberg, 2001*; *Laurino et al., 2016*), illustrate a fundamental property of these ancestral peptides: while deterministic with respect to their own structure, they are sufficiently flexible to have given rise to topologically different folds (*Copley et al., 2001*; *Grishin, 2001*; *Krishna et al., 2006*; *Alva et al., 2008*; *Pereira and Lupas, 2018b*). In this paper, we attempt to track some of this structural versatility in one of the fragments we retraced, a four-stranded β-meander found in β-propellers, β-prisms, and in the luminal domain of IRE1 (*Kopec and Lupas, 2013*).

The simplest way for peptides to achieve an increase in complexity is self-association, either by homo-oligomerization, or by amplification into a repetitive array on a single polypeptide chain (*Eck and Dayhoff, 1966*; *McLachlan, 1972*; *McLachlan, 1987*; *Remmert et al., 2010*; *Smock et al., 2016*; *Zhu et al., 2016*; *Franklin et al., 2018*; *Pereira and Lupas, 2018a*). Many folds, spanning all levels of complexity (fibrous, solenoid, toroid, and globular), clearly originated in this way. β-Propellers offer an attractive model system to study this process (*Nikkhah et al., 2006*; *Yadid and Tawfik, 2007*; *Chaudhuri et al., 2008*; *Yadid et al., 2010*; *Yadid and Tawfik, 2011*; *Voet et al., 2014*; *Smock et al., 2016*; *Noguchi et al., 2019*). They are toroids formed by sequential supersecondary structure units of typically 40–50 residues, each consisting of four anti-parallel β-strands. The units are arranged radially, with their strands perpendicular to the plane of the toroid, and have a right-handed twist, hence the name 'propeller' for the fold and 'blades' for the repetitive units. Often up to three strands of the last blade are circularly permuted to the N-terminus of the propeller, resulting in a 'velcro' closure *via* hydrogen bonding of the N- and C-terminal strands within the same blade (see *Figure 1B*), an arrangement that is thought to provide increased stability and folding specificity (*Neer and Smith, 1996*). Although all blades are structurally very similar, propellers can differ considerably in their number of blades, with structures of between four and twelve blades currently deposited in the Protein Data Bank (PDB). Also, uniquely among toroids, propellers span the full range of internal symmetry, from near sequence identity of their blades to full differentiation, where the similarity of the blades is essentially only recognizable from their structure. Diversity in sequence and structure extends to diversity in function; propellers can have enzymatic activity, bind ligands, or mediate protein–protein interactions. Often, they perform their function as a domain in the context of a larger protein.

Using bioinformatic and phylogenetic tools, we have previously shown that new propellers arise from single blades by amplification and differentiation, and that the common ancestor of the major propeller lineages was a single blade, not a fully formed propeller (*Chaudhuri et al., 2008*; *Kopec and Lupas, 2013*). This finding raises questions about the mechanisms which led to the diversity of propellers observed today. If new propellers are constantly amplified from single blades, must a given blade be amplified to the correct repeat number before it yields a folded structure, or is flexibility in copy number acceptable? Also, can incompletely amplified sequences, that is polypeptides which may be unstable on their own, fold as oligomers, allowing them to act as intermediates in the amplification process? Obviously, a certain robustness in folding would facilitate the emergence of new propellers. Several studies have addressed these questions using constructs derived from natural propellers. The first was conducted by *Nikkhah et al. (2006)*, who used a consensus WD40 blade to generate proteins with between 4 and 10 repeats, but none of the constructs appeared folded or oligomeric. Subsequently, groups in Israel and Japan built folded, oligomeric propellers from the 5-bladed tachylectin-2 propeller and from a 6-bladed NHL propeller, respectively (*Yadid et al., 2010*; *Voet et al., 2014*; *Smock et al., 2016*), but all their constructs reproduced the parental fold, suggesting that the number of blades in the final structure is not flexible, although it can be reached through oligomeric intermediates.

In parallel with the aforementioned studies, we took the same path to study the amplification of WD40 propellers. We built constructs of one- to six-bladed building blocks, derived from two seven-bladed parents with high internal sequence symmetry. In contrast to the other studies, none of our constructs reproduced the parental structure, instead yielding a diversity of forms, in an extreme case even giving rise to an arguably novel fold. Their high-resolution structures reveal the geometrical adjustments that the repeat units have to undergo to enable this structural flexibility.

## Results

### A first approach using PkwA

For our study, we decided to start from a recently amplified β-propeller, which we reasoned would not have undergone extensive differentiation towards a specific structural form. At the start of the project, the WD40 propeller with the highest internal sequence symmetry was the C-terminal domain of the putative serine/threonine protein kinase PkwA from the thermophilic actinobacterium *Thermomonospora curvata* (GenBank: AAB05822.1), whose blades mostly share >60% pairwise identity (*Figure 1A*), unreported in any propeller before (*Janda et al., 1996*). We constructed fragments

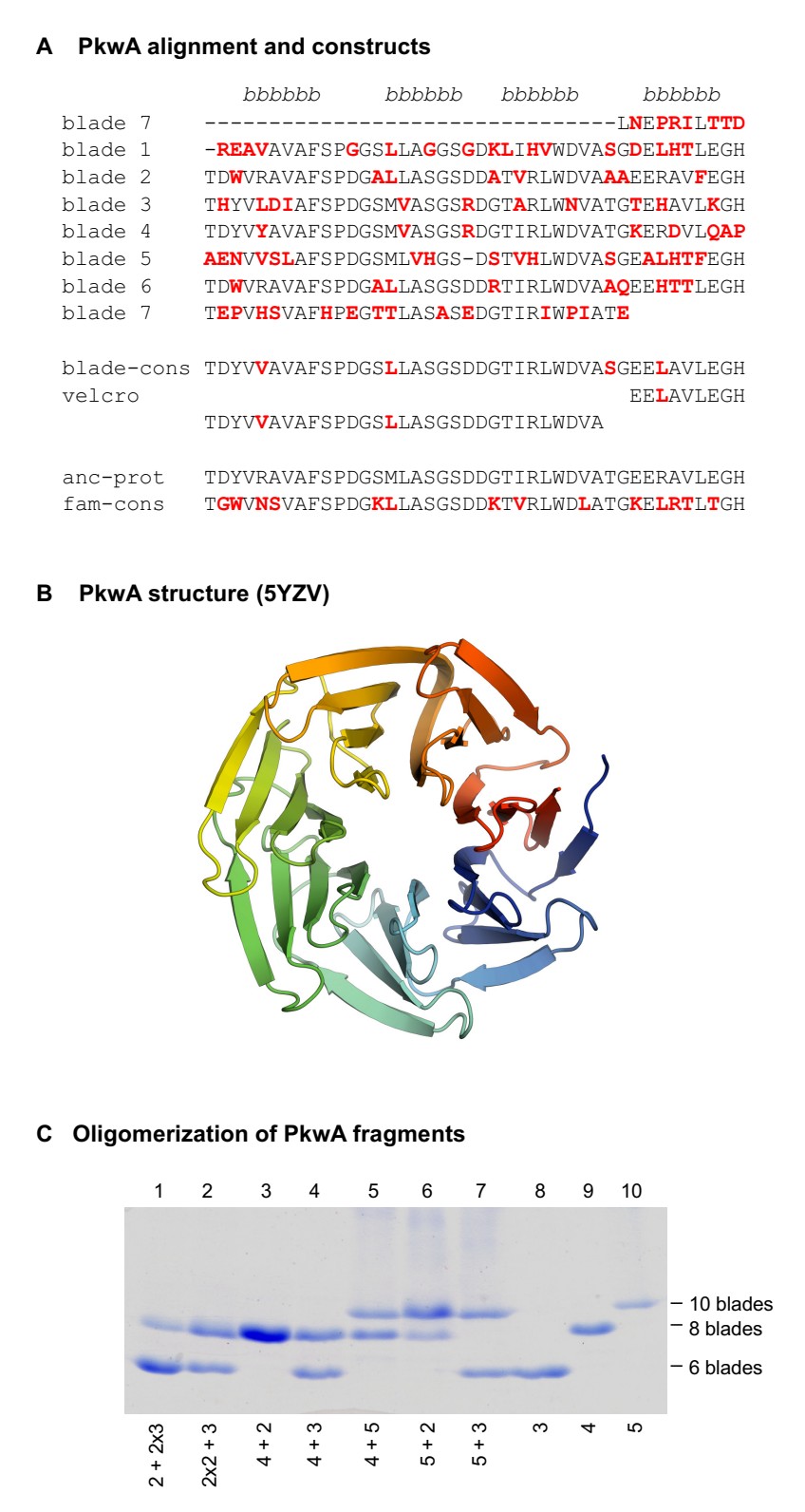

**A    PkwA alignment and constructs**

```
               bbbbbb    bbbbbb   bbbbbb    bbbbbb
  blade 7   ------------------------------LNEPRILTTD
  blade 1   -REAVAVAFSPGGSLLAGGSGDKLIHVWDVASGDELHTLEGH
  blade 2   TDWVRAVAFSPDGALLASGSDDATVRLWDVAAAEERAVFEGH
  blade 3   THYVLDIAFSPDGSMVASGSRDGTARLWNVATGTEHAVLKGH
  blade 4   TDYVYAVAFSPDGSMVASGSRDGTIRLWDVATGKERDVLQAP
  blade 5   AENVVSLAFSPDGSMLVHGS-DSTVHLWDVASGEALHTFEGH
  blade 6   TDWVRAVAFSPDGALLASGSDDRTIRLWDVAAQEEHTTLEGH
  blade 7   TEPVHSVAFHPEGTTLASASEDGTIRIWPIATE

  blade-cons TDYVVAVAFSPDGSLLASGSDDGTIRLWDVASGEELAVLEGH
  velcro                                  EELAVLEGH
             TDYVVAVAFSPDGSLLASGSDDGTIRLWDVA

  anc-prot   TDYVRAVAFSPDGSMLASGSDDGTIRLWDVATGEERAVLEGH
  fam-cons   TGWVNSVAFSPDGKLLASGSDDKTVRLWDLATGKELRTLTGH
```

**B    PkwA structure (5YZV)**

**C    Oligomerization of PkwA fragments**

**Figure 1.** The 7-bladed propeller in the protein PkwA of *Thermomonospora curvata*. (**A**) Multiple sequence alignment of the seven blades and sequences of the constructs used to test the formation of higher-order oligomers in vitro. Non-identical residues in the repeats are colored red. The four β−strands of the propeller blades are indicated above the alignment. (**B**) Crystal structure of the PkwA propeller (PDB 5YZV). The structure is
*Figure 1 continued on next page*

*Figure 1 continued*

colored in rainbow colors from blue at the N-terminus to red at the C-terminus. The velcro closure resulting from the last strand of the last blade being permuted to the N-terminus is clearly visible. (**C**) Oligomerization of PkwA consensus repeats. Differentiation of propeller sizes was achieved by native polyacrylamide gel electrophoresis. Lanes 8–10 show migration of homo-oligomeric propeller complexes assembled from 3-, 4- and 5-bladed repeats. Lanes 1–7 show mixtures of different building blocks to probe for hetero-oligomeric assembly. Proteins were mixed in equimolar ratios (lanes 3–7), unfolded and refolded together. For mixtures of 2- and 3-bladed repeats (lanes 1 and 2) 2:1 molar ratios were used. In all cases, regardless of the mixture composition, PkwA repeats re-assembled only into homo-oligomers.

DOI: https://doi.org/10.7554/eLife.49853.002

The following figure supplement is available for figure 1:

**Figure supplement 1.** Sequences of PkwA constructs.

DOI: https://doi.org/10.7554/eLife.49853.003

containing one to six tandem repeats of a PkwA consensus blade derived from a multiple sequence alignment (MSA) of its seven blades (*Figure 1A*, blade-cons; *Figure 1—figure supplement 1*). We recombinantly expressed the constructs in *E. coli* and obtained expression for all but the six-bladed construct. By far-UV circular dichroism (CD), tryptophan fluorescence spectroscopy, static light scattering (SLS), and native polyacrylamide gel electrophoresis (PAGE), we observed that the fragment comprising merely a single blade was an unfolded monomer, whereas all other fragments appeared to form folded homo-oligomers (*Table 1* upper panel; *Figure 1C*). The 2-bladed construct formed tetramers, suggesting an 8-bladed propeller and the 3-bladed, 4-bladed, and 5-bladed constructs formed dimers, suggesting the formation of 6-bladed, 8-bladed, and 10-bladed propellers, respectively. We were, however, unable to obtain high-resolution structures for any of these oligomers, evidence which we considered essential in order to judge whether such structural diversity had indeed been obtained.

For comparison, in the case of the 5-bladed tachylectin propeller, fragments with two blades reproduced the structure of the parent by forming pentamers, which assembled into two 5-bladed propellers (*Yadid et al., 2010*). Similarly, all constructs derived from a symmetrized version of the 6-bladed NHL propeller PknD reproduced the structure of the parent by assembling into higher-order oligomers (*Voet et al., 2014*). In contrast, none of our assemblies appear to have taken this path to reproduce the parental structure. We therefore mixed constructs with different numbers of blades in an attempt to generate hetero-oligomers with seven blades, but were unable to obtain any, either by co-expression or in vitro mixtures of purified components in defined stoichiometries. The fragments only reassembled efficiently into the original homo-oligomers (*Figure 1C*).

**Table 1.** Summary of biophysical data for the different propeller constructs of PkwA (upper panel) and WRAP (lower panel).

| Propeller Blades in protomer | Molecular mass protomer calculated | Molecular mass SLS measured | Assembly state based on SLS | CD melting temperature $T_m$ | Tryptophan fluorescence $\lambda_{max}$ |
|---|---|---|---|---|---|
| 2 | 8.8 kDa | 33.9 kDa | Tetramer | 52°C | 331 nm |
| 3 | 13.6 kDa | 27.8 kDa | Dimer | 67°C | 335 nm |
| 4 | 17.5 kDa | 32.7 kDa | Dimer | 63°C | 332 nm |
| 5 | 21.8 kDa | 42.6 kDa | Dimer | 65°C | 333 nm |
| 2 | 8.9 kDa | 28.4 kDa | Tetramer | 43°C | 345 nm |
| 3 | 13.2 kDa | 39.5 kDa | Trimer | 54°C | 341 nm |
| 4 | 17.6 kDa | 26.7 kDa | Dimer | 65°C | 341 nm |
| 5 | 22 kDa | 46.9 kDa | Dimer | 62°C | 341 nm |
| 6 | 26.3 kDa | 107 kDa | Tetramer | 63°C | 340 nm |

DOI: https://doi.org/10.7554/eLife.49853.004

## Foldability of single propeller blades

To investigate whether the inability of the single-bladed PkwA construct to fold and assemble into higher-order oligomers represents an exception, rather than the rule for the WD40 family, we tested several other single-bladed variants derived from PkwA as well as from an almost perfectly symmetric WD40 propeller.

In addition to the aforementioned PkwA consensus blade, we recombinantly expressed three further PkwA constructs: (a) a consensus blade derived from an MSA of PkwA homologs (*Figure 1A*, fam-cons); (b) a hypothetical ancestor of the PkwA blades computed using ancestral sequence reconstruction (anc-prot); and (c) a circularly permuted version of the PkwA consensus blade (velcro). By comparison to the non-permuted constructs, we expected the permuted one to be more amenable to folding and oligomerization. However, under all tested conditions, the three single-bladed PkwA constructs behaved as monomers with spectra characteristic of unfolded or partially folded peptides.

Since none of our single-bladed PkwA constructs were folded, we turned our attention to another highly symmetric β-propeller of the WD40 family, found in the C-terminal part of Npun_R6612 from the multicellular cyanobacterium *Nostoc punctiforme* PCC 73102 and referred to henceforth as WRAP (WD40-family Recently Amplified Propeller; ACC84870.1). WRAP, which we identified during a systematic bioinformatic survey of β-propellers (*Chaudhuri et al., 2008*), was particularly interesting in that its 14 blades are practically identical (*Figure 2A*), having acquired only 10 point mutations over 563 residues (6 of these being Trp → Arg mutations at the same position of blades 1, 3, 6, 10, 12, and 13), as a result of 13 non-synonymous substitutions at the DNA level (*Dunin-Horkawicz et al., 2014*). The 14 blades assemble into two 7-bladed β-propellers (PDB: 2YMU; *Figure 2B*). We recombinantly expressed one blade of WRAP (residues 670–710) but yet again, as for our other single-blade constructs, obtained no folding.

## Assembly of oligomeric propellers from WRAP blades

The WRAP domain was clearly amplified extremely recently, as judged from the near identity of its constituent blades and the absence of any synonymous mutations in the DNA encoding them (*Dunin-Horkawicz et al., 2014*). This domain therefore was as close to a natural domain immediately after amplification as we were likely to find, and thus precluded the need for consensus sequences and ancestral sequence reconstruction. We therefore used the aforementioned single WRAP blade to probe the foldability of fragments with tandem repeats of two to six blades (see *Figure 2—figure supplement 1* for sequences). All constructs were successfully expressed in *E. coli* and could be purified for further analysis. In CD experiments, the complexes unfolded thermally in a cooperative fashion, indicative of well-folded proteins, and their tryptophan fluorescence spectra also resembled those of folded proteins (*Table 1*).

The 2-bladed construct behaved as a single species in solution, but size determination by SLS and analytical gel size exclusion chromatography did not allow us to unequivocally distinguish between a trimeric and tetrameric form (*Table 1*). Moreover, as all WRAP fragments tended to dissociate during native polyacrylamide electrophoresis, we could not use this as an additional criterion for assessing the state of the assembly, as we had done for the PkwA constructs. Instead, we used glutaraldehyde crosslinking to assay the oligomerization state of the assemblies (*Figure 2C*), and confirmed tetramerization for the 2-bladed construct. For the 3-bladed construct, SLS data indicated a trimer, as also suggested by our crosslinking data (*Figure 2C*). This result is in contrast to the aforementioned 3-bladed PkwA construct which formed dimers, pointing to a certain versatility in the way different fragments with the same repeat number can assemble. The 4- and 5-bladed constructs assembled into homo-dimers and the 6-bladed construct into homo-tetramers. Notably, none of these fragments showed monomer-oligomer equilibria or existed in more than one form in solution (see for example the size-exclusion elution profiles in *Figure 2—figure supplement 2*), indicating that the observed assemblies were the most favorable energetically. Although there are numerous examples of monomeric 4-, 5-, and 6-bladed propellers in nature, our WRAP constructs with these blade numbers all formed homo-oligomers. Like for the PkwA constructs, we were unable to obtain hetero-oligomers by co-expression of the constructs, co-refolding of denatured proteins in vitro or co-incubation of purified components.

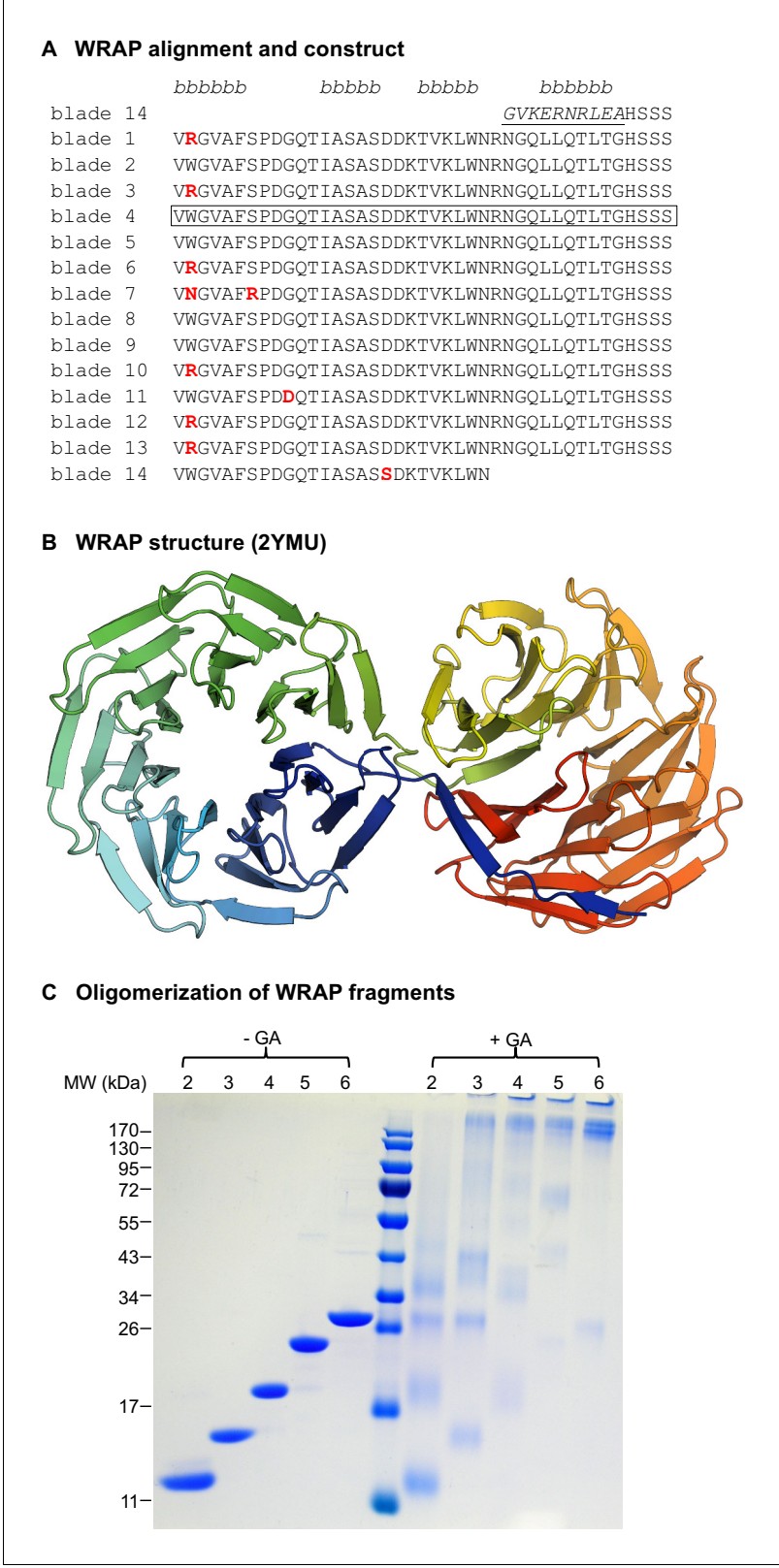

**A  WRAP alignment and construct**

```
              bbbbbb       bbbbb    bbbbb      bbbbbb
blade 14                                   GVKERNRLEAHSSS
blade 1    VRGVAFSPDGQTIASASDDKTVKLWNRNGQLLQTLTGHSSS
blade 2    VWGVAFSPDGQTIASASDDKTVKLWNRNGQLLQTLTGHSSS
blade 3    VRGVAFSPDGQTIASASDDKTVKLWNRNGQLLQTLTGHSSS
blade 4    VWGVAFSPDGQTIASASDDKTVKLWNRNGQLLQTLTGHSSS
blade 5    VWGVAFSPDGQTIASASDDKTVKLWNRNGQLLQTLTGHSSS
blade 6    VRGVAFSPDGQTIASASDDKTVKLWNRNGQLLQTLTGHSSS
blade 7    VNGVAFRPDGQTIASASDDKTVKLWNRNGQLLQTLTGHSSS
blade 8    VWGVAFSPDGQTIASASDDKTVKLWNRNGQLLQTLTGHSSS
blade 9    VWGVAFSPDGQTIASASDDKTVKLWNRNGQLLQTLTGHSSS
blade 10   VRGVAFSPDGQTIASASDDKTVKLWNRNGQLLQTLTGHSSS
blade 11   VWGVAFSPDDQTIASASDDKTVKLWNRNGQLLQTLTGHSSS
blade 12   VRGVAFSPDGQTIASASDDKTVKLWNRNGQLLQTLTGHSSS
blade 13   VRGVAFSPDGQTIASASDDKTVKLWNRNGQLLQTLTGHSSS
blade 14   VWGVAFSPDGQTIASASSDKTVKLWN
```

**B  WRAP structure (2YMU)**

**C  Oligomerization of WRAP fragments**

**Figure 2.** The recently amplified WRAP propeller in Npun_R6612 of *Nostoc punctiforme* PCC73102. (**A**) Multiple sequence alignment of the 14 blades of WRAP. Non-identical repeats are colored in red and the non-repeating β-strand 4 of the velcro blade is underlined. The four β-strands of the propeller blades are indicated above the alignment. The repeat unit chosen for in vitro studies is highlighted by a box. (**B**) Crystal structure of the WRAP

*Figure 2 continued on next page*

*Figure 2 continued*

propeller (PDB 2YMU). (**C**) Oligomerization of WRAP repeats. Assembly was probed by crosslinking proteins with 0.6% glutaraldehyde (GA) and subsequent analysis by SDS-PAGE and Coomassie Blue G250 staining. On the left side, non-crosslinked proteins are shown for comparison.

DOI: https://doi.org/10.7554/eLife.49853.005

The following figure supplements are available for figure 2:

**Figure supplement 1.** Sequences of WRAP constructs.

DOI: https://doi.org/10.7554/eLife.49853.006

**Figure supplement 2.** Purification of WRAP fragments.

DOI: https://doi.org/10.7554/eLife.49853.007

## The structure of oligomeric WRAP propellers

Unlike the PkwA constructs, many WRAP constructs crystallized and we obtained structures for homo-oligomeric assemblies of 2-, 3-, 4-, and 5-bladed fragments. As we anticipated from their oligomeric states, the 2-bladed and 4-bladed fragments formed 8-bladed propellers (*Figure 3A and B*) and the 3-bladed fragment a 9-bladed propeller (*Figure 3C*). The 5-bladed fragment, however, did not form a 10-bladed propeller, even though it was dimeric, as observed in solution. Rather, the dimer adopted an unusual, asymmetric structure, hitherto unseen in natural proteins (*Figure 3D*), in which one monomer formed the expected 5 blades of an incomplete propeller, while the other monomer formed only four blades and converted part of the N-terminal fifth blade to a helix, leaving the rest of this blade unstructured. The blades in one monomer were antiparallel to the blades in the other, as opposed to natural propellers, where the blades are always parallel. Thus, while the fold is clearly built of blades, it is not a propeller. As the stretch of sequence preceding the helix was not resolved in the structure, we analyzed the intactness of the crystallized protein and found that it had undergone partial trimming during crystallization (*Figure 3—figure supplement 1*).

In highly symmetrical propellers, the interface between blades is specific for the number of blades in the overall structure. However, this is not what we observe in our assemblies. Instead, the packing of the blades within each fragment preserves the geometry of the 7-bladed parent (*Figure 3E*), and the tension due to the mismatch between the number of blades in the assembly versus that in the parent accumulates at the interface between the fragments (*Figure 3F*). In the 9-bladed propeller, the symmetry mismatch is absorbed to equal extent at three interfaces, leading to a slightly triangular shape. In the two 8-bladed propellers, the one formed by four 2-bladed fragments absorbs the symmetry mismatch to different extent at the four interfaces, leading to a slightly oval shape and an approximate two-fold symmetry. The one formed by two 4-bladed fragments absorbs the tension at only two interfaces, leading to an even more pronounced oval shape and a clear two-fold symmetry. This latter construct, unlike all others, also shows considerable interface variability between the propellers in the asymmetric unit of the crystal, resulting from the two halves being shifted to different extent in the plane of the interface, further highlighting the inherent structural tension. This is presumably also the reason why the 5-bladed fragment does not form a 10-bladed propeller: The structural tension resulting from the mismatch between the 7-bladed parent and a 10-bladed assembly is apparently too large to be absorbed at two fragment interfaces. Instead, one monomer changes orientation, improving the geometry at one of the two interfaces, and compensates for the accumulated tension at the other interface by conversion of its N-terminal blade to a helix, thus also reducing to nine the number of blades that have to be accommodated in the structure.

## Discussion

We have explored the structural versatility of propeller blades, which represent one of the ancestral peptides predating folded proteins (*Alva et al., 2015*). For this we used two symmetrical β-propellers of the WD40 family, one of which has clearly been amplified extremely recently in evolution and naturally shows an almost perfect internal symmetry (*Chaudhuri et al., 2008*; *Dunin-Horkawicz et al., 2014*), obviating the need to bring it about computationally (*Voet et al., 2014*; *Noguchi et al., 2019*). Constructs derived from this very highly symmetrical protein, WRAP, yielded two oligomeric propellers with eight blades and one with nine blades, as well as a new fold, which departs from the propeller architecture. None of the constructs recapitulated the 7-bladed structure

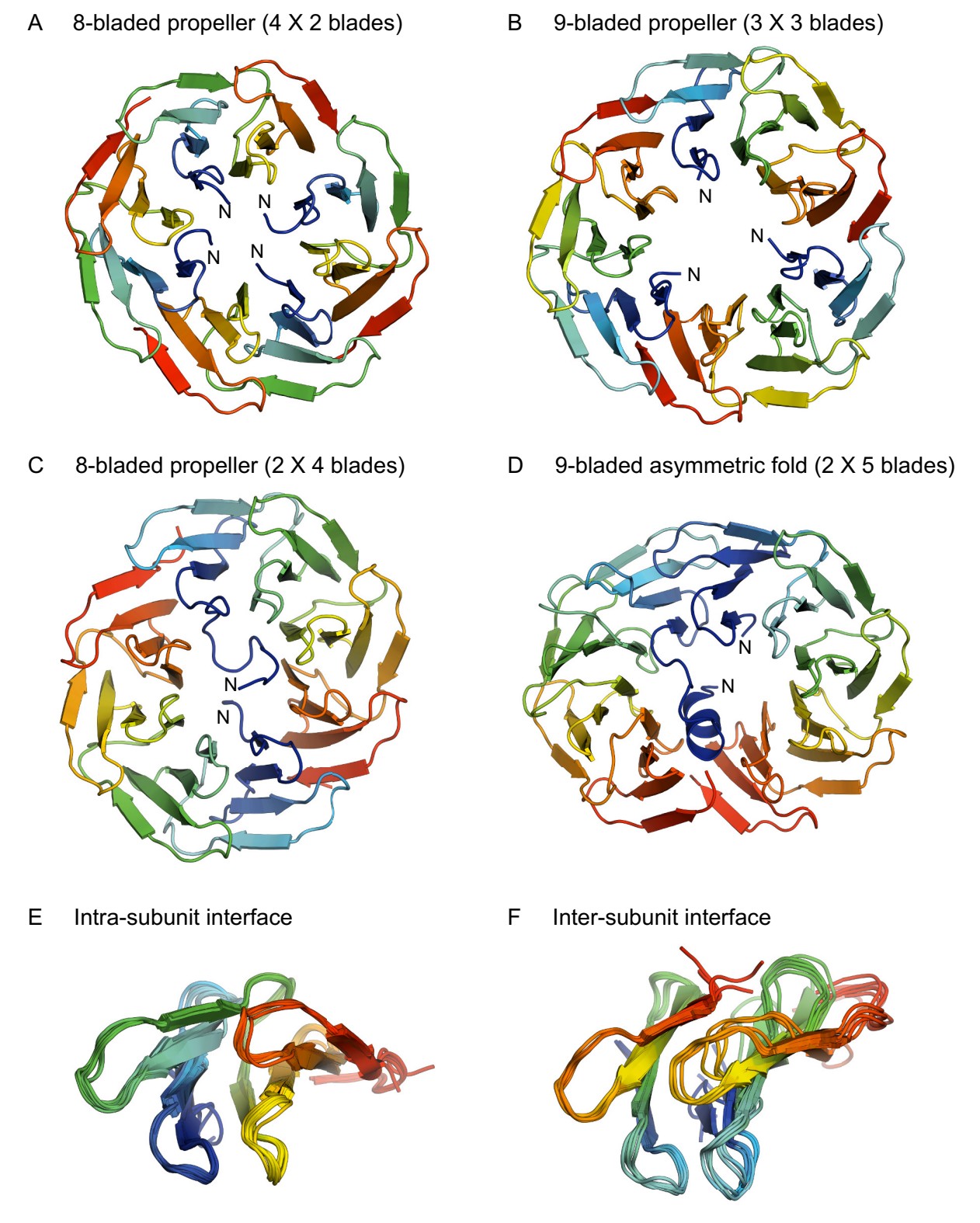

**Figure 3.** Structures of homo-oligomeric WRAP propellers. Subunits in each propeller are colored in rainbow colors with blue at the N-terminus and red at the C-terminus. (A) 8-bladed propeller formed of four 2-bladed fragments (PDB 6R5X). (B) 9-bladed propeller formed of three 3-bladed fragments (6R5Z). (C) 8-bladed propeller formed of two 4-bladed fragments (6R5Y). (D) 9-bladed asymmetric fold formed of two 5-bladed fragments

*Figure 3 continued on next page*

*Figure 3 continued*

(6R60). (**E**) Superimposition of all intra-subunit interfaces in 2-, 3-, 4-, and 5-bladed fragments. (**F**) Superimposition of all inter-subunit interfaces in the two 8-bladed propellers and the 9-bladed propeller.

DOI: https://doi.org/10.7554/eLife.49853.008

The following figure supplement is available for figure 3:

**Figure supplement 1.** Truncation of 5-bladed WRAP repeats correlates with an asymmetric, incomplete propeller structure.

DOI: https://doi.org/10.7554/eLife.49853.009

of the parent, in contrast to previous efforts based on the 5-bladed tachylectin-2 propeller (*Yadid and Tawfik, 2007*; *Yadid et al., 2010*; *Smock et al., 2016*) and on a 6-bladed NHL propeller (*Voet et al., 2014*), in both of which all constructs assumed the parental fold. Our results thus show that the same blade can form propellers of different symmetry when amplified to different copy numbers within one polypeptide chain. This supports the notion that subdomain-sized protein fragments are more flexible structurally than fully formed folds – and thus available for reuse in the formation of new proteins (*Nepomnyachiy et al., 2017*; *Alva and Lupas, 2018*) – particularly if the fragments have not yet had the time to evolve specificity for a particular structure.

Given the small sample size of propeller constructs attempted so far, it is of course impossible to judge whether the WD40 family is intrinsically more versatile than other families, but we note that the first set of WD40 constructs we made, based on the PkwA consensus blade, also showed structural versatility. While it is not clear in the absence of high-resolution data whether the tetramer of 2-bladed constructs and dimers of 4- and of 5-bladed constructs indeed formed the expected structures, or rather assumed the parental 7-bladed architecture with one or more blades left unstructured, the dimer of 3-bladed constructs could not have reached the parental architecture and would thus most likely have formed a 6-bladed propeller. Provided that the PkwA-derived constructs did form the expected structures, this blade would thus potentially have been even more versatile, since the WRAP constructs did not produce any 6-bladed form. A counterpoint to the view that WD40 blades may be particularly versatile is however provided by recent results with a computationally symmetrized 8-bladed WD40 propeller, Tako8 (*Noguchi et al., 2019*). A version with seven blades, Tako7, could not be expressed and one with nine blades, Tako9, formed an 8-bladed structure with a single unfolded blade.

An intriguing, open question concerns the 5-bladed PkwA construct, which assembles into a dimer. Is this dimer a 10-bladed propeller, a 7-bladed propeller with substantial unstructured extensions, or a 'hopeful monster' with the same fold as the one formed by the 5-bladed WRAP dimer? The concept of hopeful monsters was introduced by the geneticist Richard Goldschmidt in order to describe the sudden emergence of new species through discontinuous variation at the genetic level (*Goldschmidt, 1940*). While the subject of ongoing controversy in developmental biology, this concept is quite attractive in order to describe the relationship between the new fold and canonical propeller structures. The change in its genetic information is not large, going from 2, 3, 4, or seven repeat units to 5, but it is certainly discontinuous and leads to the sudden emergence of a new form, which – as a stable, efficiently folded protein – is available for further evolution. A corollary of these considerations is therefore the question whether this new fold can be functionalized sufficiently to allow it to enter biological evolution in vivo. Our efforts in this direction have just started. We note however that, unless the WRAP blade has unique properties not clear to us, nature must have seen proteins with this fold on many occasions along the eons.

## Materials and methods

### Bioinformatics

As a starting point for the consensus constructs we used the propeller (residues 450–740) of PkwA from *Thermonospora curvata* (AAB05822.1), annotated as a putative serine/threonine-protein kinase. The blades in this propeller are on average 60% identical and their alignment, shown in *Figure 1*, was used to derive the majority rule blade consensus for PkwA. To obtain a majority rule family consensus, we searched the non-redundant protein database from NCBI using BLAST (*Altschul et al., 1997*) for sequences similar to PkwA at a significance level better than 1e-3 and a minimal coverage

**Table 2.** Crystallization conditions and cryo protection

| Construct | Protein solution | Reservoir solution (RS) | Cryo solution |
|-----------|------------------|--------------------------|---------------|
| *2-blades* | 8 mg/ml protein<br>50 mM TRIS HCl pH 8.0<br>150 mM sodium chloride | 200 mM sodium acetate<br>100 mM TRIS HCl pH 8.5<br>30%(w/v) PEG 4000 | n/a |
| *3-blades* | 23 mg/ml protein<br>50 mM TRIS HCl pH 7.5<br>150 mM sodium chloride | 200 mM ammonium fluoride<br>20%(w/v) PEG 3350 | RS + 10%(v/v) PEG 400 |
| *4-blades* | 23 mg/ml protein<br>50 mM TRIS HCl pH 7.5<br>150 mM sodium chloride | 10 mM zinc chloride<br>100 mM Hepes pH 7.0<br>20%(w/v) PEG 6000 | RS + 10%(v/v) PEG 400 |
| *5-blades* | 4 mg/ml protein<br>50 mM HEPES pH 7.5<br>100 mM sodium chloride | 100 mM Magnesium chloride<br>100 mM HEPES pH 7.0<br>15%(w/v) PEG 4000 | RS + 15%(v/v) PEG 400 |

DOI: https://doi.org/10.7554/eLife.49853.010

of 80%. To reconstruct a possible ancestral blade, we used the individual blades of PkwA and applied the Ancescon software (*Cai et al., 2004*), with the marginal reconstruction method, alignment-based rate factor, and alignment-based PI vector.

## Protein cloning and expression

The coding DNA sequences for the various single blades were constructed from two overlapping oligonucleotides each, which were complemented in a DNA polymerase reaction (Pfu polymerase, Stratagene). The DNA fragments were cloned into the vector pGEX4T1 (GE Healthcare) with BamHI/XhoI restriction sites, leading to Glutathion-S-transferase (GST) fusion proteins with a thrombin cleavage site N-terminally to the respective single blade. To obtain constructs with two or more blades containing repetitive sequences of the PkwA consensus, additional DNA fragments coding for these blades were fused together by PCR with the help of a linker oligonucleotide. The 2-bladed construct was cloned into vector pET-30b (Novagen) using NdeI/HindIII restriction sites, whereas the 5- bladed propeller fragment was cloned into pET-28b to generate a construct with an N-terminal cleavable His$_6$-tag. The 3- and 4-bladed constructs were expressed as GST fusion proteins, while the 6-bladed construct was made as a GST fusion as well as in a His$_6$-tagged form. Genes encoding 1–6 repeats of WRAP protomers were synthesized (Eurofins) and cloned into pETM-11 using NcoI/XhoI restriction sites, generating constructs with an N-terminal His$_6$-tag cleavable by TEV-protease. Correctness of all clones was confirmed by DNA sequencing. Recombinant plasmids were transformed into *E. coli* strains C41(DE3) and BL21-Gold (DE3) grown on agar plates containing 50 µg/ml kanamycin (pET vectors) or 100 µg/ml ampicillin (pGEX4T1). For expression, cells were cultured at 25°C in LB medium with the respective antibiotic and induced with 1 mM isopropyl-D-thiogalactopyranoside (IPTG) at an OD$_{600}$ of 0.6 for continued growth overnight.

## Protein purification

Bacterial cell pellets were resuspended in phosphate buffered saline (PBS), supplemented with DNaseI (Applichem) and protease inhibitor cocktail (Complete, Roche). After breaking the cells in a French Press, the suspension was centrifuged twice at 37,000 g. For GST-fusion proteins, the supernatant was bound to a glutathione affinity column (GSTrap FF, GE Healthcare) in buffer A (50 mM Tris pH 8.0, 50 mM NaCl) and eluted in buffer A with 10 mM reduced GSH. The eluted protein was adjusted to buffer B (20 mM Tris, 300 mM NaCl, 1 mM CaCl$_2$, 1 mM dithiothreitol, pH 8.5) and incubated with 1 U thrombin/mg protein at room temperature for 2–6 hr. The cleaved protein was dialysed against PBS and re-applied to a GSTrap FF column, from which the flow-through, containing the cleaved blades, was concentrated and purified by gel size exclusion chromatography (HiLoad 26/60 Superdex G75, GE Healthcare) in buffer C (50 mM Tris pH 7.5, 50 mM NaCl). Due to the thrombin cleavage site sequence, all proteins expressed as GST-fusions start with additional Gly-Ser residues (*Figure 1—figure supplement 1*).

For purification of the 2-bladed PkwA consensus repeat, protein supernatant was bound to a phenyl sepharose column (High Load FF 16/10, GE Healthcare). The column was washed with buffer C

**Table 3.** Crystallographic data collection and refinement statistics

| Construct (PDB ID) | 2-blades (6R5X) | 3-blades (6R5Z) | 4-blades (6R5Y) | 5-blades (6R60) |
|---|---|---|---|---|
| Data collection | | | | |
| Space group | C222$_1$ | P2$_1$ | P2$_1$2$_1$2 | C2 |
| Cell dimensions | | | | |
| a, b, c (Å) | 55.24, 119.7, 84.15 | 53.55, 92.35, 61.43 | 97.72, 127.2, 72.96 | 39.75, 107.5, 179.2 |
| α, β, γ (°) | 90.00, 90.00, 90.00 | 90.00, 94.95, 90.00 | 90.00, 90.00, 90.00 | 90.00, 94.40, 90.00 |
| Resolution (Å) | 32.3–1.70 (1.80–1.70) * | 38.3–1.75 (1.85–1.75) * | 38.7–2.15 (2.28–2.15) * | 39.8–1.75 (1.85–1.75) * |
| $R_{merge}$ | 4.8 (56.7) | 6.3 (89.4) | 11.0 (76.4) | 8.8 (45.4) |
| I / σI | 17.8 (2.32) | 13.5 (1.55) | 10.1 (1.94) | 9.17 (1.92) |
| Completeness (%) | 99.3 (97.4) | 99.2 (95.9) | 99.4 (99.0) | 98.1 (95.3) |
| Redundancy | 4.31 (4.38) | 4.67 (4.44) | 3.70 (3.51) | 3.31 (3.40) |
| Refinement | | | | |
| Resolution (Å) | 32.3–1.70 | 38.3–1.75 | 38.7–2.15 | 39.8–1.75 |
| No. reflections | 29426 | 56974 | 47400 | 70952 |
| $R_{work}$/$R_{free}$ | 0.20/0.24 | 0.19/0.21 | 0.22/0.25 | 0.20/0.24 |
| No. atoms | | | | |
| Protein | 2364 | 5357 | 7350 | 5639 |
| Ligands (Zn$^{2+}$) | 0 | 0 | 6 | 0 |
| Water | 314 | 302 | 330 | 691 |
| B-factors | | | | |
| Protein | 24.30 | 32.30 | 36.50 | 26.70 |
| Ligands (Zn$^{2+}$) | - | - | 50.60 | - |
| Water | 35.30 | 36.70 | 33.70 | 35.50 |
| R.m.s. deviations | | | | |
| Bond lengths (Å) | 0.012 | 0.017 | 0.011 | 0.013 |
| Bond angles (°) | 1.55 | 1.72 | 1.51 | 1.53 |

*Values in parentheses are for highest-resolution shell.

DOI: https://doi.org/10.7554/eLife.49853.011

and protein eluted with 30% ethanol in water. After 1:10 dilution with buffer C, the protein was bound to an anion exchange column (MonoQ 16/10, GE Healthcare) and eluted with a linear salt gradient up to 1 M NaCl. The final purification step consisted of gel size exclusion chromatography (Superdex G75 26/60, GE Healthcare) in PBS.

His$_6$-tagged proteins were purified by binding proteins to Ni-NTA columns (GE Healthcare) in buffer D (50 mM Tris pH 8.0, 300 mM NaCl) and elution with increasing concentrations of imidazol up to 0.6 M. Eluted proteins were dialyzed against buffer B for cleavage by thrombin (1 U/mg protein) or against buffer A for cleavage by TEV protease (50 U/mg protein), respectively. After incubation overnight, cleaved proteins were concentrated and finally purified by gel size exclusion chromatography (Superdex G75 26/60, GE Healthcare) in buffer C.

## Biophysical analysis

Circular dichroism (CD) spectra from 200 to 250 nm were recorded with a Jasco J-810 spectropolarimeter at room temperature in 0.5x concentrated PBS buffer. Cuvettes of 1 mm path length were used in all measurements. For melting curves, CD measurements were recorded at 208 nm from 10–95°C, the temperature change was set to 1 K per minute, using a Peltier-controlled sample holder unit. Tryptophan fluorescence spectra were recorded on a Jasco FP-6500 spectrofluorometer; excitation was at 280 nm, emission spectra were collected from 300 to 400 nm.

## Assembly of protein complexes

For assembly experiments, the proteins were mixed in the molar ratios indicated in the figure legends. Native protein mixtures were incubated for 30 min at room temperature. Unfolded protein mixtures in 6 M GdmCl were incubated at room temperature for 1 hr and dialysed against buffer D overnight. The refolded samples were analyzed by native PAGE on a 15% gel and stained using Coomassie Blue G250. Oligomerization was also probed by crosslinking; protein (0.25 µM) was incubated with 0.6% glutaraldehyde for 20 min at room temperature in buffer E (20 mM HEPES pH 7.5, 150 mM NaCl). Reactions were stopped with 0.1 M Tris pH eight and samples were analyzed by SDS-PAGE. To determine the native molecular mass of assembled protein complexes, analytical gel size-exclusion chromatography was performed with 1–2 mg protein in buffer A on a Superdex 75 10/300 column (GE Healthcare), to which a static light-scattering (SLS) detector (miniDawn, Wyatt Technologies) and a refractive index detector (RI-2031, Jasco) were coupled. Molecular masses of the analyzed proteins were determined with the Wyatt Technologies software package, AstraV v5.1.5.

## Crystallization, Data Collection, Structure Solution and Refinement

Crystallization trials were performed at 294 K in 96-well sitting-drop vapor-diffusion plates with 50 µl of reservoir solution and drops consisting of 300 nl protein solution and 300 nl reservoir solution. The composition of protein solutions and crystallization conditions for the crystals used in the diffraction experiments are listed in *Table 2* together with the solutions used for cryo-protection. Where applicable, single crystals were transferred into a droplet of cryo-solution before loop-mounting and flash-cooling in liquid nitrogen. All data were collected at beamline X10SA (PXII) at the Swiss Light Source (Paul Scherrer Institute, Villigen, Switzerland) at 100 K and a wavelength of 1 Å using a PILATUS 6M detector (DECTRIS). Diffraction images were processed and scaled using the XDS program suite (*Kabsch, 1993*).

Using MOLREP (*Vagin and Teplyakov, 2000*), the structures were solved by molecular replacement using 2-, 3-, 4- and 5-bladed fragments of the 7-bladed WRAP propellers (PDB 2YMU) as search models. All models were completed by cyclic manual modeling with Coot (*Emsley and Cowtan, 2004*) and refinement with REFMAC5 (*Murshudov et al., 1999*). Data collection and refinement statistics are summarized together with PDB accession codes in *Table 3*.

## Mass spectrometry

To probe the crystalized 5-bladed construct for proteolysis, crystals were dissolved in SDS-sample buffer and analyzed by SDS-PAGE. Excised bands were digested with AspN protease and analyzed using an ion trap Orbitrap Elite mass spectrometer (Thermo Scientific; Proteome Center, University of Tübingen).

## Acknowledgements

We thank Tancred Frickey for the calculation of the ancestral sequence of the PKWA blades, Mirita Franz-Wachtel (Proteome Center, University of Tübingen) for Mass Spectrometry analysis, Kerstin Bär and Reinhard Albrecht for assistance with the crystallization experiments, and the staff of beamline X10SA at the Swiss Light Source (PSI, Villigen, Switzerland) for excellent technical support. This work was supported by institutional funds of the Max Planck Society and by a 'Life'-Grant from the Volkswagenstiftung to ANL.

## Additional information

### Competing interests

Andrei N Lupas: Reviewing editor, *eLife*. The other authors declare that no competing interests exist.

## Funding

| Funder | Grant reference number | Author |
|---|---|---|
| Max Planck Society | | Andrei N Lupas |
| Volkswagen Foundation | Life Grant 94 810 | Andrei N Lupas |

The funders had no role in study design, data collection and interpretation, or the decision to submit the work for publication.

## Author contributions

Evgenia Afanasieva, Investigation, Cloned, purified and characterized WRAP constructs, Produced preparations for crystallization assays; Indronil Chaudhuri, Investigation, Cloned, purified and characterized PkwA constructs, Produced preparations for crystallization assays, Identified WRAP as a highly symmetrical propeller; Jörg Martin, Supervision, Validation, Methodology, Writing—original draft, Project administration; Eva Hertle, Investigation, Biophysical characterization of constructs, Produced preparations for crystallization assays; Astrid Ursinus, Investigation, Biophysical characterization of the constructs, Produced preparations for crystallization assays; Vikram Alva, Visualization, Writing—original draft, Writing—review and editing, Structure analysis; Marcus D Hartmann, Data curation, Validation, Investigation, Writing—original draft, Solved all crystal structures, Structure analysis; Andrei N Lupas, Conceptualization, Formal analysis, Supervision, Funding acquisition, Methodology, Writing—original draft, Writing—review and editing, Bioinformatic and structure analyses, Produced the final version of the manuscript

## Author ORCIDs

Vikram Alva https://orcid.org/0000-0003-1188-473X
Marcus D Hartmann https://orcid.org/0000-0001-6937-5677
Andrei N Lupas https://orcid.org/0000-0002-1959-4836

## Decision letter and Author response

Decision letter https://doi.org/10.7554/eLife.49853.021

# Additional files

## Data availability

Diffraction data have been deposited in PDB under the accession codes 6R5X, 6R5Z, 6R5Y, and 6R60.

The following datasets were generated:

| Author(s) | Year | Dataset title | Dataset URL | Database and Identifier |
|---|---|---|---|---|
| Afanasieva E, Chaudhuri I, Martin J, Hertle E, Ursinus A, Alva V, Hartmann MD, Lupas AN | 2019 | Structural diversity of oligomeric $\beta$-propellers with different numbers of identical blades | http://www.rcsb.org/structure/6R5X | Protein Data Bank, 6R5X |
| Afanasieva E, Chaudhuri I, Martin J, Hertle E, Ursinus A, Alva V, Hartmann MD, Lupas AN | 2019 | Structural diversity of oligomeric $\beta$-propellers with different numbers of identical blades | http://www.rcsb.org/structure/6R5Z | Protein Data Bank, 6R5Z |
| Afanasieva E, Chaudhuri I, Martin J, Hertle E, Ursinus A, Alva V, Hartmann MD, Lupas AN | 2019 | Structural diversity of oligomeric $\beta$-propellers with different numbers of identical blades | http://www.rcsb.org/structure/6R5Y | Protein Data Bank, 6R5Y |
| Afanasieva E, | 2019 | Structural diversity of oligomeric $\beta$- | http://www.rcsb.org/ | Protein Data Bank, |

| Chaudhuri I, Martin J, Hertle E, Ursinus A, Alva V, Hartmann MD, Lupas AN | propellers with different numbers of identical blades | structure/6R60 | 6R60 |
|---|---|---|---|

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
