## [Decision Letter]

[Editor’s note: the editors did not request any revisions prior to acceptance, so there is not an accompanying Author response.]

Congratulations, we are pleased to inform you that your article, "Structural diversity of oligomeric β-propellers with different numbers of identical blades", has been accepted for publication in *eLife*.

It is most likely that proteins evolved complex structures by multiple duplication of small building blocks. Varying in the number of blades and sequence similarity between the blades, β-propellers offer a great example to learn about such evolution. In this work, the authors study the ability of β-propeller "blades" to reassemble into higher-order structures when allowed to do so separately from their parent structures. This had been attempted before, but without success in that the assembled structures simply resembled the parent protein, or did not fold.

Here, the authors chose as the parent protein a recently formed propeller fold in a protein (WRAP). The recent origin of this protein is deduced from the high internal similarity (near identity) between its 7 blades within each of two propeller domains, pointing to evolutionarily recent fusion and divergence from the parental blade.

The authors took a sequence of a blade from a 7-bladed propeller and investigated how proteins composed of its repeats (from 2 to 6) fold. It was interesting to learn that it was unexpectedly difficult to make stable proteins from identical repeats, and quite a few resisted 3D structure determination. Those that were successful mostly produced symmetric structures through oligomerization. Due to this apparent preference for symmetry probably stemming from energetically favorable interactions between the blades within each monomer, none of the structures reproduced their 7-bladed progenitor, but only grew in the number of blades.

It was interesting to see that the 6-bladed structure never formed, that is, why didn't 3-bladed chains assemble into 6-bladed dimers? However, 8 or 9-bladed propellers were obtained. The most unexpected result was the 5-bladed chain. It didn't assemble into a 10-bladed dimer. Why? Is it because the sequence in the original 7-bladed unit was such that going beyond 9 creates steric clashes? It also didn't assemble into a symmetric 9 or 8-bladed unit by kicking out one or two blades from one or both chains. Instead, it formed a bizarre asymmetric assembly in which the two chains in a dimer packed in antiparallel fashion. Only 9 blades were accommodated, and one blade of one chain was indeed kicked out of this broken propeller that couldn't close due to antiparallel arrangement of the chains.

These experiments revealed that such new and unusual folds can originate rather easily and may form the basis of structure inventions, like those that seen in viruses. The reviewers found work quite fascinating. Going forward, it would be interesting to learn how these structures could evolve, either for stability or for function, if strict identity of repeats is relaxed (in future work, of course, not in this paper). E.g., what path would that 9-bladed asymmetric monster take upon some random mutations being introduced in its sequence. Will it continue to be asymmetric, or it will relax back to maybe 10-bladed "normal" dimer? Novel folds are probably easy to get by wrecking up the structure somewhat, being saved by oligomerization. But would they stay novel in evolution?

The work is very well done technically, and the results are solid. We recommend publication.